# Influence of Circadian Rhythm in the Eye: Significance of Melatonin in Glaucoma

**DOI:** 10.3390/biom11030340

**Published:** 2021-02-24

**Authors:** Alejandro Martínez-Águila, Alba Martín-Gil, Carlos Carpena-Torres, Cristina Pastrana, Gonzalo Carracedo

**Affiliations:** 1Ocupharm Research Group, Biochemistry and Molecular Biology IV, Faculty of Optics and Optometry, 28037 Madrid, Spain; amarting@ucm.es (A.M.-G.); ccarpena@ucm.es (C.C.-T.); crispast@ucm.es (C.P.); jgcarrac@ucm.es (G.C.); 2Research & Development Department Avizor, 28919 Leganés, Spain; 3Optometry and Vision Department, Complutense University, 28037 Madrid, Spain

**Keywords:** circadian rhythm, ocular diseases, glaucoma, melatonin, melanopsin, myopia, dry eye

## Abstract

Circadian rhythm and the molecules involved in it, such as melanopsin and melatonin, play an important role in the eye to regulate the homeostasis and even to treat some ocular conditions. As a result, many ocular pathologies like dry eye, corneal wound healing, cataracts, myopia, retinal diseases, and glaucoma are affected by this cycle. This review will summarize the current scientific literature about the influence of circadian patterns on the eye, focusing on its relationship with increased intraocular pressure (IOP) fluctuations and glaucoma. Regarding treatments, two ways should be studied: the first one, to analyze if some treatments could improve their effect on the ocular disease when their posology is established in function of circadian patterns, and the second one, to evaluate new drugs to treat eye pathologies related to the circadian rhythm, as it has been stated with melatonin or its analogs, that not only could be used as the main treatment but as coadjutant, improving the circadian pattern or its antioxidant and antiangiogenic properties.

## 1. Circadian Rhythm in the Eye

The term circadian rhythms (circa meaning ‘‘around”, and dian meaning ‘‘day”) was first coined by Halberg in 1953 [1] and refers to biological cycles with periods slightly longer than 24 h (on average ~24.2 h) [2]. Body temperature, sleep duration, hormonal levels, heart rate, and other physiological variables exhibit such daily oscillations [3,4].

In mammals, there are peripheral circadian clocks located in tissues like retina, heart, liver, lungs, pituitary, and skeletal muscles that contain their own circadian oscillators [5,6]. However, the suprachiasmatic nucleus (SCN) of the anterior hypothalamus is considered to be the master circadian clock since its discovery in 1970s [7,8].

The way to synchronize this rhythm is by external cues known as “zeitgebers” (‘‘time givers” in German). Some of these zeitgebers are temperature, feeding times, and social interactions. Nevertheless, the primary and most important cue is light, which allows us to synchronize SCN to the day–night cycle.

Through the eye, light reaches the retina and is processed by retinal ganglion cells (RGCs), which drive whole visual information. However, approximately 1–2% of these cells contain a photopigment denominated melanopsin [9,10]. This small subset of RGCs received the name of intrinsic photosensitive retinal ganglion cells (ipRGCs) which are directly photosensitive [11,12]. This photic input is carried to the SCN via the retinohypothalamic tract using glutamate as a neurotransmitter which also acts in gene expression, which is important in the circadian process [13]. Moreover, ipRGCs project their axons to the intergeniculate leaflet [14] and to the olivary pretectal nucleus of the pretectum to mediate pupil light reflex [15,16].

### 1.1. Melanopsin

It was in 1998 when the photopigment melanopsin was first discovered in the eye and skin of a frog (*Xenopus laevis*) [17]. In mammals, however, it has been only localized in retina, encoded in the gene OPn4. This opsin has a maximum peak of absorption at approximately 482 nm [12]. This property allows ipRGCs to act as additional photoreceptors to the classical cones and rods, processing non-image-forming information [18].

Implications of melanopsin in the nonvisual information are supported by studies in mice without cones and rods that still responded to light stimulus with pupillary constriction [19], melatonin suppression [20], and phase shift. Additional studies in mice with the melanopsin gene ablated only in ipRGCs showed deficits in circadian photoentrainment and pupillary light reflex [21,22,23]. Furthermore, blind people with retinal damage (cones and rods) have normal photoentrainment and pupil responses [24].

However, melanopsin is also responsible for an important process, the melatonin synthesis suppression. Its activation by short-wavelength light (470–480 nm) [25,26] decreases melatonin levels both in the central nervous system (CNS) as in blood and in the eye, following a circadian rhythm although inversed between them. 

### 1.2. Melatonin

Melatonin (N-acetyl-5-methoxytryptamine) is the hormone responsible for regulation of circadian and seasonal rhythms [27,28]. This molecule was first discovered and described in the pineal gland [29], but currently it is known to be synthesized in many tissues in the body including the eye and ocular annexes, specifically in the retina [30], iris, ciliary body [31], crystalline lens [32,33], and lacrimal gland [34], where it regulates important processes. As mentioned, melatonin synthesis is controlled by light–darkness cycles, increased during the night and suppressed during the day [35], reaching a concentration peak at night (between 02:00 to 04:00) [36]. It is initiated from tryptophan in pinealocytes [29,37], then converted to 5-hydroxytryptophan which, in turn, is transformed into serotonin (by tryptophan 5-hydroxylase and aromatic amino acid decarboxylase, respectively). Following that, serotonin is converted to N-acetylserotonin (NAT) by arylalkylamine N-acetyltransferase (AANAT) and posteriori to melatonin by hydroxyindole-0-methyltransferase (HIOMT) (Figure 1).

Melatonin action is mediated by cellular receptors. Two types of membrane melatonin receptors have been characterized and cloned in mammals: MT_1_ and MT_2_, both being G-protein-coupled receptors with seven transmembrane domains [38,39]. A third putative receptor called MT_3_ [40] was characterized as the enzyme quinone reductase 2 [40,41], having lower affinity for melatonin when compared to MT_1_ and MT_2._

MT_1_ and MT_2_ receptors are expressed in different parts of the brain including SCN, cerebellum, hippocampus, substantia nigra, ventral tegmental area, in cardiovascular system, and in almost all peripheral organs: blood vessels, adrenal glands, mammary gland, gastrointestinal tract, liver, kidney and bladder, ovary, testis, prostate, skin, and the immune system [42].

Focusing on the eye, receptors have been localized in different structures. On the ocular surface, melatonin MT_1_ receptor has been detected in corneal epithelial cells in different animals [43,44] and humans [45], while MT_2_ receptor has only been identified in animals [43,46,47]. They have also been seen in rabbits in the iris and ciliary body [48] and in human uveal melanocytes and human uveal melanoma cells [49]. Presence of melatonin in aqueous humour [50] has also been confirmed as its enzymes AANAT and HIOMT [51]. Furthermore, within the lens, MT_2_ receptor has been identified [43] being considered as another source of melatonin [52]. In sclera, both receptors are expressed [43,44]. 

In the posterior segment, melatonin is also synthesized in the retina by photoreceptors, and its receptors are localized in many areas of the retina, with the MT_2_ receptor being more expressed than MT_1_ in human eyes [38]. These receptors have been detected in plexiform layers, horizontal cells, amacrine cells, ganglion cells, and in rod photoreceptors cells [42,43,45,53,54,55,56]. Accordingly, melatonin seems to play a role modulating dopamine release [57], in the regulation of retinomotor movements [58], visual sensitivity [59], and mediate regeneration of photoreceptors external discs [60].

Synthesized melatonin is released into circulation as an endocrine hormone and binds to all receptors of target organs to regulate various physiological processes associated [55]. Melatonin levels can be measured in blood, saliva, and urine samples [61], and to keep a normal circadian rhythm, exposure to light–dark cycles should follow a regular pattern. 

Nowadays, artificial lights are present almost everywhere, especially in electronic devices. This fact can affect our biological rhythms and is currently the subject of numerous studies. Increase in exposition to lights at evening and at night can delay or suppress normal melatonin secretion [62], through the activation of melanopsin contained in ipRGCs [18,63,64,65]. Intensity and duration of the exposure, as well as light spectrum, are considered variables in this process. Currently, there are many devices using LED light sources with a short-wavelength emission thought to be responsible for the increase in sleep disorders [66].

In general, circadian disruption has been linked not only to sleep disorders or jet lag but also to different conditions such as obesity, diabetes, cardiovascular disease, cancer, psychiatric disorders, and neurodegenerative diseases [67]. Moreover, it represents a common feature of Alzheimer [68,69] and Parkinson diseases [70,71,72], showing a strong decrease in melatonin levels [73,74,75]. Additionally, some studies carried out with shift workers, whose circadian rhythm is altered, have shown a high prevalence of some of the aforementioned diseases [76,77,78].

This review will highlight the ocular pathologies such as dry eye, cataract, myopia, retinal diseases, and glaucoma which are affected by this rhythm, focusing its relationship with IOP fluctuations and glaucoma, and how melatonin and its analogs modify IOP.

## 2. Ocular Pathologies Related to Circadian Rhythm 

In addition to the implications of the eye in the regulation of systemic circadian rhythm, a large number of ocular physiological processes are also controlled by it. Any biochemical or optical mechanisms involved in the circadian cycle could lead to developing ocular pathologies or alterations of ocular physiology. In this section, we have reviewed the role of circadian rhythm and melatonin (Table 1) in some ocular conditions such as dry eye, corneal wound healing, myopia, cataract, and retinal diseases. Glaucoma will be more deeply analyzed in the following sections.

### 2.1. Dry Eye

Dry eye is a multifactorial disease of the ocular surface characterized by a loss of homeostasis in the tear film [96]. Due to the ocular-surface health status being directly dependent on several environmental factors [97], it is difficult to establish if the diurnal/nocturnal variations in the ocular-surface physiology are related to these environmental factors, the circadian rhythm, or a combination of both.

Regarding tear volume, its diurnal variations have been confirmed, which would follow a circadian cycle [98,99,100]. In healthy subjects, there is a discrepancy between studies reporting higher tear volume during the morning [98,100] and others during the afternoon [99], but they all showed lower tear volume late in the day. On the other hand, tear osmolarity also seems to follow a circadian pattern [99,101], despite a study of Oncel et al. [102] showing a stable profile throughout the daytime. Niimi et al. [101] demonstrated that tear osmolarity was drastically decreased upon awakening, quickly increased until stabilizing its value during the first 8 h, and finally slightly increased before sleep, unlike tear volume [98,99,100]. In the case of dry-eye patients, their circadian rhythm of both tear volume and tear osmolarity could be altered compared to healthy subjects [99,100]. 

Supporting the idea that dry eye could be influenced by the circadian rhythm, the relationship between sleep disorders and dry-eye signs and symptoms has been proposed [103]. In mice, Li et al. [104] demonstrated that sleep deprivation induced dry eye, affecting different parameters such as tear secretion, ocular-surface damage, lacrimal composition, or producing lacrimal-gland hypertrophy. In humans, different studies showed that the prevalence of sleep disorders is higher in dry-eye patients [105,106,107] and how these sleep alterations exacerbate its signs and symptoms [108]. 

Concerning the significance of melatonin in dry eye, it is necessary to point out that the presence of this molecule in human tears has been recently described, following a circadian cycle with a higher concentration during the night [109]. In rabbits, a study performed by Navarro Gil et al. [79] discovered the secretagogue effect of melatonin analog, agomelatine, on tear secretion, converting them into therapeutic candidates for dry eye. Conversely, the topical instillation of melatonin did not produce an effect on tear secretion. Nevertheless, an earlier study by Hoyle et al. [80] showed that the combination of melatonin with diadenosine tetraphosphate (Ap_4_A), another tear secretagogue, produced a synergic effect by increasing tear secretion.

### 2.2. Corneal Wound Healing

Corneal wound healing is a physiological process that involves a series of events of cell migration, proliferation, adhesion, and differentiation. This process presents a high interest in different ocular-surface pathologies, including dry eye [110]. It is known that both the renewal of corneal epithelium and its regeneration is regulated by circadian rhythm [111,112,113]. However, it is not clear when the maximum cell mitosis is produced, due to the differences found amongst animal species. For example, Sasaki et al. [114] found a higher mitotic rhythm in the corneal epithelium during the night in Japanese quail (diurnal animal), while Xue et al. [115] found this peak during the morning in mice (nocturnal animal). Under this context, it is reasonable to think that maximum mitotic activity in humans would be produced during the night, but this affirmation needs to be further tested in future studies.

The fact that corneal wound healing depends on circadian rhythm allowed the elucidation of using melatonin to accelerate this process. Crooke et al. [47] studied the effect of topical instillation of melatonin and its analogs in a corneal-wound model in rabbits. There was an improvement of around 47% in the rate of healing with both melatonin and IIK7, while 5-MCA-NAT showed no effect. The reverse effect of melatonin due to its combination with luzindole (nonselective melatonin antagonist) and DH97 (MT_2_ receptors antagonists) confirmed that MT_2_ receptor mediates the accelerating effect of melatonin. Additionally, Crespo-Moral et al. [81] recently developed an ex vivo corneal-wound model in porcine eyeballs that allowed us to confirm the accelerating effect of melatonin during the first 72 h of the healing process.

### 2.3. Myopia

Besides myopia being a distance vision impairment when it is uncorrected, high levels of myopia (≥6.00 diopters) produce structural ocular changes increasing the risk of developing pathologies such as myopic maculopathy, glaucoma, or cataracts [116]. For this reason, the control of myopia progression in the age of ocular growth is essential to prevent this condition from becoming pathological [117].

Axial length, choroidal thickness, and different molecules involved in eye growth have been demonstrated to be regulated by circadian rhythm. Concerning axial length, diurnal and nocturnal animals present opposite rhythms [118,119]. In humans, axial length is longest during the day and decreases during the night, and it seems that there are no differences between emmetropic and myopic subjects [120,121]. In the case of the choroidal thickness, its circadian cycle is opposed to axial length [121,122], which would confirm that ocular growth reduces the thickness of this vascular layer. On the other hand, in vitro studies demonstrated that the synthesis of scleral proteoglycans [123], responsible for the structural properties of the sclera, is higher during the morning, and it is also increased in myopic chicks [124,125]. These findings are aligned with a recent study performed in humans by Read et al. [126], who found an increase in the anterior scleral thickness during the night compared to its diurnal values.

In 1993, Weiss and Schaeffel [118] reported the relationship between myopia development and circadian rhythm in chicks for the first time. They found that axial growth cycles were altered when myopia was induced by depriving the eye of form vision, showing a constant growth during day and night. Since then, many clock-related genes and neurotransmitters have been discovered to be involved in myopia development, including melanopsin, melatonin, and its MT_1_ receptor [118,127,128,129].

After that, it was possible to discover that children spending more time outdoors and exposed to increased light have not only a lower prevalence of myopia but also lower myopic changes in both nonmyopic and myopic children [130,131,132,133,134,135,136]. The effect of ambient light in myopia control has been confirmed in animal studies, where it was found that the presence of dopamine plays an important role in reducing myopia progression [129,137,138,139]. It should be considered that dopamine and melatonin circadian cycles oscillate in antiphase in the central nervous system [140], including the retina [141]. In this sense, recent studies showed that melatonin levels in body fluids such as serum and saliva were elevated in myopic subjects compared to emmetropes ones [142,143], but they were reduced in urine [144]. In accordance with the above, dopamine levels were also reduced in the serum of these myopic subjects [142].

The idea of using light to control the progression of myopia was established at the beginning of this century. Several authors confirmed the efficacy of retinal stimulation with short-wavelength light (blue) to reduce the eye growth in chicks [145,146,147] and guinea pigs [82,148,149,150], in opposition to mid- (green) and long-wavelength (red) light. Wang et al. [82] demonstrated that blue light (480 nm) specifically stimulates synthesis of melanopsin in retina and sclera, reducing both expression of melatonin MT_1_ receptor and concentration of melatonin in the pineal gland. These findings concur with a recent study in guinea pigs performed by Zheng et al. [83], who studied the effect of melanopsin inhibition by intravitreal injection of an antagonist (AA92593). This melanopsin inhibition produced an abnormal increase in eye growth that was directly correlated to increased melatonin levels in the retina.

### 2.4. Cataracts

Cataract formation is a physiological process caused by the aggregation of fiber cells in the lens, producing a yellow opacification due to alterations in crystallins, the main structural protein [151]. Despite cataracts being an age-related process affecting vision, excessive light exposure also leads to oxidative damage of the lens, which accelerates its opacification [152].

Considering that a cataract acts as a natural yellow filter that blocks blue light, it is logical to consider its influence on circadian rhythm. On the one hand, localization of the MT_1_ receptor in lens epithelial cells [43] has been suggested as being responsible for circadian regulation of their mitosis [153,154]. Moreover, the presence of melanopsin in human lens epithelium was recently described by Alkozi et al. [52], who found that its specific stimulation with blue light (465–480 nm) reduces melatonin levels in the lens, while darkness, green light (520–550 nm), and red light (625–640 nm) increase these levels. On the other hand, concerning sleep disorders that could derive from the lens yellowing, a meta-analysis performed by Erichsen et al. [155] concluded that cataract surgery could help to improve them.

As mentioned above, oxidative stress plays an important role in cataract formation. Therefore, due to the antioxidant properties of melatonin, this molecule has been presented as a therapeutic candidate to prevent opacification. In rat models, several authors discovered how the use of melatonin reduced lipid peroxidation and promoted both synthesis of glutathione, an antioxidant, and antioxidative activity of different enzymes, leading to a reduction of cataract formation [84,85,86,87,88,89,90]. Additionally, regulation of melatonin release in the lens could open a door to new ocular therapies. In this respect, Pintor [91] found that the inhibition of melanopsin present in the lens of rabbits with a yellow filter (absorbance at 465–480 nm) and its antagonist AA92593 reduced concentration of ATP in aqueous humour.

### 2.5. Retinal Diseases

The retina is not only the principal ocular structure which synthesizes melatonin, but also some of its physiological processes are regulated by circadian rhythm. These processes include the regulation of photoreceptor sensitivity to light, renewal of their outer segments, and phagocytosis of fragments of these segments by retinal pigment epithelium cells, as well as gene expression or production of melatonin and dopamine [156]. In diurnal animals, the synthesis of melatonin receptors of different retinal layers also follows a circadian cycle. Melatonin receptors increase under light conditions and decrease under darkness, unlike melatonin levels [44,56,157]. On the other hand, the levels of vascular endothelial growth factor (VEGF) secreted by retinal pigment epithelium regulating the angiogenesis are increased under daylight conditions [158]. Concerning the preclinical effect of melatonin, a study by Sugawara et al. [159] found that the systemic administration of luzindole, a nonselective melatonin receptor antagonist, protected photoreceptors from light-induced damage in rats. Contradicting these results and in line with other authors, Liang et al. [92] showed that the systemic administration of melatonin also reduced photoreceptor damage and apoptosis in a mouse retinal degeneration model. This contradiction between the beneficial and harmful effects of melatonin would manifest the complex role of circadian rhythm in retinal neuroprotection. Nevertheless, antioxidant and antiangiogenic properties of melatonin make it an excellent candidate for treatment of AMD and diabetic retinopathy.

Concerning the therapeutic use of melatonin in AMD, different studies confirmed its antioxidant and antiangiogenic properties, which provide a protective effect on photoreceptors and retinal pigment epithelium against oxidative stress, apoptosis, and mitochondrial DNA damage [94,160,161,162]. 

Due to diabetic retinopathy also being accompanied by retinal oxidative stress and angiogenesis, therapeutic use of melatonin has been proposed for this disease [93,160,161,162]. It has been demonstrated that synthesis of melatonin in the retina is reduced in diabetic rats [163], while, in diabetic patients with proliferative retinopathy, concentration of this molecule in serum is also reduced during nighttime [164]. Several studies performed on animal models demonstrated the protective effect of melatonin against retinal inflammation, oxidative stress, angiogenesis, and apoptosis, which could help to prevent retinal damage associated with diabetes [165,166]. Nevertheless, the clinical benefits of melatonin are still unknown due to the lack of clinical studies.

## 3. Relation of Circadian Rhythm with IOP Fluctuations and Glaucoma

Glaucoma is a multifactorial optic neuropathy characterized by the damage of the optic nerve head and irreversible loss of vision [167]. Epidemiology studies estimated that glaucoma is the leading cause of irreversible blindness worldwide, affecting approximately 76 million individuals aged 40–80 years in 2020 [168]. Its pathophysiology is not fully understood, but it seems that a mechanical stress and a reduction in the retinal blood flow are related to gradual damage of the retina, causing damage of the progressive retinal ganglion cells (RGCs), dysfunction, and death, starting from the periphery toward the center of the retina [169].

Increased intraocular pressure (IOP) is a major risk factor in the developing of this pathology [170,171,172]. For this reason, most of the pharmacological and surgical treatments of glaucoma typically involve lowering IOP, either by decreasing the production of aqueous humour (AH) or by increasing its outflow [173,174]. Normal IOP (~16 mmHg in humans) is the hydrostatic pressure within the eye which is generated by the correct balance between production and drainage of aqueous humour, and it is essential to permit normal vision [175]. The production of AH involves two consecutive processes. Firstly, a portion of plasma from ciliary processes is filtered through the fenestrated capillaries into the interstitial space between vessels and ciliary epithelia. Then, a portion filtered is actively secreted into the posterior chamber by ciliary epithelial cells. In humans, these processes are affected by age, circadian rhythm, and the presence of glaucoma [176]. The rate of AH formation in human drops from 2.6 µL/min during daytime to 1.1 µL/min at night [177]. This rate of production decreases with age, by approximately 15–35% between ages 20 and 80 years [178]. Its rate of formation is significantly lower at night in healthy subjects and in untreated glaucoma patients [179,180], but glaucoma patients have a higher rate of AH formation at night [179]. On the other hand, the majority of aqueous humour (70–90%) in human eyes drains through the “conventional” outflow pathway in a pressure-dependent pattern, leaving the anterior chamber through trabecular meshwork and Schlemm’s canal in internal scleral sulcus from iridocorneal angle. A minor proportion (10–30%) of aqueous humour drains via “non-conventional” aqueous outflow or uveoscleral pathways, which is not pressure-dependent, through the sclera via intercellular space between ciliary muscle fibers and its connective tissue [181]. Outflow facility is decreased at night in older healthy subjects [182], reducing with age [183,184], in parallel with age-related reduction in aqueous-humour production [185]. Nevertheless, outflow facility in ocular hypertensive and glaucoma patients is significantly lower than in healthy subjects [179,184]. 

### 3.1. Circadian Rhythm of IOP and OPP

In light of this, it is clear why IOP is variable and influenced by circadian rhythm. IOP fluctuates in healthy subjects within a range of 4–6 mmHg in the course of a day [186,187,188], with maximum values at daybreak and minimum values at the end of afternoon [189,190]. In patients with ocular hypertension (OHT), mean described fluctuation is 6–8 mmHg, with a mean increase of 15 mmHg during peak IOP [191,192]. While, in glaucoma patients, IOP fluctuates in a range between 6 and 15 mmHg, with a maximum limit of up to 40 mmHg in extreme cases [185,187]. Greater IOP fluctuation increase the odds of visual field loss progression by 30% [193]. Most 24 h IOP monitoring studies describe higher readings during morning, between 6:00 a.m. and 12:00 p.m. (Figure 2) [194,195,196]. This matches with the aqueous-humour production and outflow pattern that significantly diminishes during sleep [185]. However, some authors suggest that better, well-designed studies on the importance of circadian IOP fluctuation are needed, because there is still some controversy with respect to its basis behavior [176]. For example, Liu et al. [189] reported that a true circadian IOP rhythm, aside from posture-related changes (sited while awake, lower than supine while asleep), is responsible for a nocturnal IOP increase in healthy adults due to an increased outflow resistance at night instead of a nocturnal IOP decrease as described above. They described that IOP troughs appeared at 9:30 p.m. which is a peak at the end of the night till before awakening (approximately at 5:30 a.m.) [197]. Moreover, in a subsequent study, they found the average increase in IOP from day to night was higher in younger healthy subjects (5.1 mmHg) compared to older healthy individuals (4.5 mmHg), demonstrating that IOP is elevated at night and during supine sleep in non-glaucomatous individuals [198]. However, in this sense, Mansouri et al. [199] caution that age differences need to be considered when IOP curves from different studies are compared. They found that older, healthy subjects had a mean peak IOP at around 10:20 a.m., whereas the peak for younger healthy individuals was earlier, between 5:30 a.m. and 6:30 a.m. depending on body posture. Furthermore, this peak also depends on the glaucomatous condition, occurring during day, rather than at night [180] for normal-tension glaucoma patients and primary open-angle glaucoma (POAG) eyes [200] while afternoon peaks were more common in primary chronic-angle-closure glaucoma eyes after a laser iridotomy, as occurs in normal eyes. 

Additionally, other factors are noted to affect glaucoma onset and progression such as systemic blood pressure [189] and ocular perfusion pressure (OPP) [190], which also follow circadian patterns. In this sense, blood pressure follows a distinctive circadian curve characterized by systolic and diastolic decline during sleep, with a trough roughly between 2:00 a.m. and 4:00 a.m. [189,191]. This decrease is followed by a transient peak in arterial pressure at mid-morning [192]. In this sense, Graham and Drance [193] showed that patients with greater nocturnal blood pressure decrease were more likely to have shown visual field loss progression despite seemingly adequate IOP control. Furthermore, in the early manifest glaucoma trial (EMGT), Leske et al. [194] found certain vascular risk factors were predictors of glaucoma progression being both systemic hypertension [195] and hypotension [196] potential risk factors for glaucoma. Regarding ocular perfusion pressure (OPP), it is defined as two-thirds of mean blood pressure minus IOP [197], and it has been proposed by World Glaucoma Association as a risk factor in POAG [176]. In this sense, Liu et al. described a OPP peak which occurred at night for both younger and older subjects [198]. OPP and diurnal-to-nocturnal increase in IOP is greater in older subjects throughout 24 h, so aging may compromise autoregulatory capacity normally present in healthy younger individuals that ensures adequate ocular blood flow [176]. Moreover, in glaucoma patients, OPP is lower than control, being at its minimum around 7:00 a.m., just before awakening [197]. For its part, Choi et al. noted that wider circadian OPP fluctuations in glaucoma patients were associated with excessive nocturnal blood pressure decrease and with worse visual field indices [199], as a risk factor for glaucoma [200]. Other authors state that circadian OPP fluctuation is the most consistent clinical risk factor for glaucoma progression severity [201]. They found that both anatomic (retinal nerve fiber layer thickness) and functional (visual field) outcome variables were significantly worse in glaucoma patients with wider circadian OPP fluctuation. This suggests that an associated glaucoma progression to eyes with defective autoregulatory mechanisms that could not maintain consistent OPP [202]. 

### 3.2. Circadian Rhythm of Melatonin and Glaucoma

Taking all this into account, it is clear that aqueous-humour flow and IOP are influenced by circadian rhythm. As has been described previously in the text, melatonin is one of the hormones responsible for the regulation of circadian and seasonal rhythms. In addition, it is known that melatonin is normally elevated during the night in the blood serum and in the eye. It has been described that in young healthy subjects, the melatonin serum peak is about 65 pg/mL (67.32 ± 9.18 pg/mL), and it appears around 2.30 h, while in elder individuals (>70 years old) melatonin serum peak is lower than younger individuals, 29.2 ± 6.1 pg/mL [203]. Nevertheless, melatonin levels may be significantly altered in eye pathologies for instance in glaucoma [204]. Studies performed by Alkozi et al. report that nonglaucomatous patients who exhibit IOP higher than 21 mmHg presented concomitantly higher concentrations of melatonin [50]. While normotensive patients (IOP below 21 mmHg) presented values of 14.6 ng/mL in aqueous humour, hypertensive patients showed melatonin concentrations of 46.6 ng/mL. These results are in agreement with Ma et al. [204], whose previous studies measuring melatonin levels in blood in the morning (measurements taken from 7 a.m. to 10 a.m.) found higher serum concentrations in glaucoma patients compared with the control group. Thereby, several studies have correlated diurnal changes in IOP and fluctuation of melatonin levels and have suggested that some melatoninergic mechanism is involved in circadian rhythm of intraocular pressure [205,206]. In this sense, studies performed on animal models [207] and more recently on humans [208,209,210] suggest that melatonin or any of its analogs can significantly reduce IOP. Furthermore, the mechanism by which melatonin produces a reduction of IOP is described in detail in the section below.

As previously stated, it is currently unclear whether circadian issues affecting melatonin secretion lowers IOP, or whether ocular hypertension, a known risk factor for glaucoma, causes melatonin-related circadian issues. Both pathways are probably not independent, and both may contribute to further progression of glaucoma. 

As described above, glaucoma causes a significant retinal ganglion cell loss, including intrinsically photosensitive retinal ganglion cells (ipRGCs) [211]. Drouyer et al. [212] found that glaucoma caused an average 72% reduction of ipRGC innervations to SCN compared with control group. In consequence, glaucoma may be the main ophthalmic disease affecting the progression of circadian-rhythm issue. This ipRGCs loss may decrease light perception and thus may result in abnormal diurnal variation of melatonin secretion, furthering systemic circadian-rhythm issue. Studies performed in human glaucomatous patients showed excess amounts of melatonin secreted during atypical time frames [204]. Therefore, the relationship between glaucoma and melatonin seems to be bidirectional, as glaucoma inducing the death of ipRGC may affect rhythm of pineal melatonin production, which in turn, may affect circadian system activity, as well as abnormal melatonin secreted which could be involved in the pathogenesis of glaucoma. However, it is still unclear which mechanism arises first. 

On the other hand, despite this background, glaucoma has been reported to be associated with a high incidence of sleep disorders [213,214,215], seasonal affective disorder or depression, and anxiety [216]. Some circadian rhythm abnormalities may induce sleep and mood disorders as a result of loss in vision due to glaucoma progression [215,217]. Exactly how the mechanism of glaucoma may induce sleep problems is not clear, but one possibility is that the ipRGC loss that occurs in glaucoma results in decreased light inputs to the SCN via retino-hypothalamic tract. This may result in a decreased synchronization of the SCN with the 24 h circadian rhythm, altering neurotransmitters in the cerebrum by means of irregular secretions of cortisol, norepinephrine, acetylcholine, or melatonin throughout the day, thereby resulting in sleep disorders [218]. 

In parallel, glaucoma patients with visual fields defects were four times as likely to receive depression scores > 9 on the Beck depression inventory [219]. The range of prevalence of depression in glaucoma patients was between 11.4 and 32.1% in an Australian clinical study performed by Skalicky et al. [220]. Several studies have reported that nocturnal disturbance of melatonin secretion may be related with the pathogenesis of major depressive disorders [221,222,223]. It may be due to melatonin being able to inhibit adrenocorticotropic hormone-stimulated cortisol production [224]; therefore, an abnormally reduction in melatonin secretion may be related with the hypercortisolemia described as one of the most prevalence signs in depressive individuals [225]. However, further studies are necessary to fully characterize the relationship between glaucoma, circadian rhythms, and mood disorders. 

## 4. Melatonin and Analogs: Its Function on IOP

Melatonin has been described since 1984 as a natural compound with hypotensive effect [226]. Its action in IOP is mediated through MT_1_, MT_2_, and the putative MT_3_ melatonin receptor, located in the ciliary body, which leads to a decrease in chloride efflux from non-pigmented epithelial cells (NPE) through an increase in cAMP [227]. Lowering this efflux causes a decrease in aqueous-humour production and finally, a reduction of IOP. 

A significant number of melatonin analogs have also shown hypotensive efficacy in New Zealand white rabbits: N-acetyltryptamine, 6-chloromelatonin, 2-iodomelatonin, 2-phenylmelatonin, IIK7, agomelatine [207], and 5-MCA-NAT [228]. Between them, 5-MCA-NAT was the most efficacious one, reaching almost 50% IOP reduction and lasting for more than 8 h [228]. Furthermore, this reduction is not fully reverted to basal values and could last for up to 96 h [229].

Another melatonin analogue with a good lowering effect is agomelatine. This compound is a potent agonist of the MT_1_ and MT_2_ melatonin receptors [230] and a nonselective inhibitor of the 5-HT_2C_ serotonin receptor [231]. This compound is approved for the treatment of major depression [232] but when applied topically in New Zealand white rabbits, it reduces 20% IOP [207]. 

Studies with melatonin analogs have also been done on hypertensive conditions. The Trendelenburg position [233] consists of placing the animal with the head down at different angles to the horizontal causing a status venous that increases IOP from the first minute and for at least 20 min, reaching 160% (5.2 mmHg more) [207]. In such a position, melatonin, agomelatine, and 5-MCA-NAT could increase their efficacy by 50% more with respect to normotensive conditions [207], showing the higher IOP, and obtaining better results. 

Looking for a more complex animal model with fully developed glaucoma, DBA/2J mice were selected. They spontaneously develop pseudo-exfoliative glaucoma [234] increasing their IOP from 6 months reaching the glaucoma stage at 9 months with marked alteration in electroretinogram [235]. Experiments using this model have also proven that melatonin and 5-MCA-NAT were able to reduce IOP by a half more in glaucomatous mice than in control mice [236], as occurred in hypertensive conditions. Furthermore, a long-term application of 5-MCA-NAT for 3 months could partially counteract the rise in IOP observed in nontreated glaucomatous mice [236]. This treatment not only produced an acute IOP reduction but also a long-term effect with a final difference of 18% between both groups.

As previously stated, agomelatine was approved for major depression in 2009, and after our group performed experiments to confirm its effect on IOP, an interesting study was performed with hypertensive psychiatric patients. In it, agomelatine was used to treat their condition, but IOP was also evaluated, showing a reduction of roughly 30% after 15 and 30 days of treatment [208].

However, not only has agomelatine been used in humans, but also melatonin in different studies and conditions, showing promising results. Samples was the first one who used melatonin on normotensive humans to test its hypotensive effect, showing that with only 500 µg of melatonin, IOP could decrease [237]. Years later, Ismail et al. demonstrated that 10 mg melatonin before cataract surgery not only reduced IOP from 17.9 to 13.8 mmHg but also produced anxiolytic effect, leading to better surgical conditions and outcomes [209]. Recently, a nutritional supplement based on 1 mg melatonin has also been developed especially for glaucoma patients, showing that melatonin not only reduced IOP in an acute effect but could also reduce IOP in the morning during the 3 days tested [210].

It has to be pointed out that combined therapy for glaucoma control is a common method when monotherapy is no longer effective [238]. Consequently, this combination of melatonin with other hypotensive compounds could not only be beneficial in terms of IOP reduction, but also for its antioxidant and antiangiogenic properties [93,94,95,160,161,162,239] as mentioned in point 2.5 retinal diseases. Furthermore, there are some studies which show promising results about a possible slow down vision worsening over time in glaucoma. Park et al. show that RGC survival was greater in retinas treated with melatonin 2 weeks after ischemia-reperfusion [240], and Belforte et al. demonstrate that melatonin prevented and reversed the effect of ocular hypertension on retinal function and diminished the vulnerability of retinal ganglion cells during a 6 week period [206].

Although melatonin receptor activation is the main cause of IOP reduction [228], there are some results that cannot explain IOP reduction patterns, such as the increase in 5-MCA-NAT effect with corynanthine, an α1 antagonist [227], or when there is both hypertension and increased melatonin levels [50]. In those cases, functional units constituted by melatonin receptors in complex with other proteins must be considered. 

It has been reported that MT_1_ and MT_2_ receptors may form heteromers in a heterologous expression system [241,242]. Furthermore, MT_2_ receptor may interact with serotonin 5-HT_2C_ [243], being of particular interest due to the discover of agomelatine, a potent agonist of the MT_1_ and MT_2_ melatonin receptors [230] and a non-selective inhibitor of the 5-HT_2C_ serotonin receptor [231]. Recently, interactions of MT_1_ and MT_2_ receptors with α_1_ antagonist have been shown to increase melatonin hypotensive effect to almost 50% and lasting for 6 h [244].

Apart from the aforementioned interactions with heteromers, there are key results from melatonin and 5.MCA-NAT regarding regulation of gene expressions that impact on their interaction with other hypotensive compounds. In this way, both compounds reduce the expression of carbonic anhydrase 2 (CAII) and 12 (CAXII) in in vitro assays [229]. Moreover, in vivo results confirm this theory. A single application of 5-MCA-NAT could raise dorzolamide (a carbonic anhydrase inhibitor) IOP reduction applied for 3 days, increasing its efficacy by a half up to 45% [229].

A similar behavior has been obtained with adrenoceptors. On the one hand, melatonin and 5-MCA-NAT were able to significantly reduce the expression of the β_2_-adrenergic receptor and up-regulate the α_2A_ receptor. On the other hand, if a single dose of melatonin or 5-MCA-NAT was applied, and then daily treated with either timolol (nonselective β-adrenergic receptor antagonist) or brimonidine (selective α_2A_-adrenergic agonist), the hypotensive effect of the adrenoceptor-acting drugs was increased especially for the later, reaching almost 80% reduction [245]. 

Finally, the effect of melatonin and its analogs on different animal models or patients has been summarized in Table 2.

## 5. Summary

This review summarizes the current scientific literature on the influence of circadian patterns on the eye. The circadian rhythm is involved in many processes in different parts of the eye. From the ocular surface, the most anterior structure of the eye, to the retina, the most posterior, the circadian rhythm, and the molecules are involved in playing an important role to regulate the homeostasis and even to treat some conditions. For instance, the influence of sleep disorders on dry-eye symptoms and signs has been demonstrated [103]. Melatonin analogues were found to be a potential treatment of dry eye, improving the symptomatology and mainly signs like tear volume or tear composition [79,109]. In the case of melatonin, no effect has been found directly over the dry-eye symptoms, more than its capacity to improve corneal wound healing [47]. However, its role in sleep-disorder regulation could be used as coadjutant treatment of ocular surface diseases due to the improvement in sleep disorders. In the posterior pole of the eye, the effect of circadian rhythm is not clear but melatonin properties such as antioxidant and antiangiogenic effects could be important to enhance other main treatments for AMD or other retinal diseases [94,160,161,162]. 

Nowadays, the most significant protagonist of circadian rhythm on the eye is about the refractive development role and the IOP regulation. Probably one of the most important concerns for ophthalmology is the myopia incidence increasing in recent decades on a global basis. Some Asian countries present incidences of over 90% in children between 10 to 13 years old. Myopia is already widely recognized as a significant public health issue, causing visual loss and a risk factor for a range of other serious ocular conditions [247]. Recent studies showed that molecules involved in the circadian rhythm, and even the own circadian rhythm, play an important role in the axial growth of the eye [248]. For instance, some studies have found that the endogenous circadian rhythm in sclera modifies the scleral proteoglycan synthesis underlying the oscillations in the eye length. In addition, Weis and Schaeffel [118] found that eye elongation is higher during the day than at night, and Nickla et al. [125] described less axial growth in myopic defocus if the light, as treatment, is applied in the afternoon rather than the morning. Moreover, potential treatments for inhibiting melatonin and enhancing dopamine have been proposed [249]. It is very interesting to observe that classical treatment, such as atropine to control myopia progression, which seemed to have its effect over the accommodation to reduce the axial growth of the eye, really is controlled by a regulator of G-protein signaling (RGS). Concretely, this role is performed by RGS2, a feedback inhibitor of melatonin production in the pineal gland [250], being down-regulated in the sclera by atropine and modifying the scleral collagen layer thickening, hindering the axial growth [251]. Regarding IOP, it seems clear that it is regulated in part by circadian rhythm because aqueous-humour formation, and its outflow through the trabecular meshwork are different depending on the day or night. Moreover, for glaucoma, other factors affect glaucoma onset and progression such as systemic blood pressure [189] and ocular perfusion pressure [190], both following circadian patterns. However, these IOP changes during the day do not only affect glaucoma but probably eye growth as well. Some studies have found that changes in IOP are related to modifications in scleral proteoglycan synthesis, and therefore, it is possible that IOP rhythm influence the eye size and the refraction development [125]. On the other hand, melatonin and its analogues have been described as potential drugs for glaucoma treatment, decreasing the IOP.

## 6. Conclusions

The diagnosis of certain eye conditions could be improved if an all-day monitoring of some parameters were made. It is currently possible to analyze the IOP using a specific contact-lens sensor able to capture the IOP for 24 h [252]. This type of measurement could help to get better diagnosis and therefore to prescribe the best treatment for the patient. Regarding treatments, two ways should be studied: the first one, to analyze if some treatments could improve their effect on the ocular disease if their posology could be established in function of circadian patterns. For instance, is it better to instill atropine for myopia control progression or a hypotensive drug for glaucoma in the morning or at night? The second one, to evaluate new drugs to treat eye pathologies related to the circadian rhythm, as melatonin or its analogues, not only as the main treatment but as coadjutant, improving the circadian pattern or for its antioxidant and antiangiogenic properties.

## Figures and Tables

**Figure 1 biomolecules-11-00340-f001:**
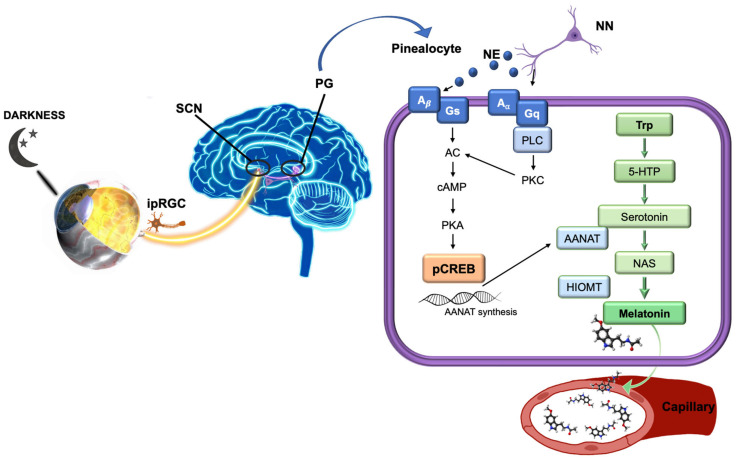
Diagram of melatonin synthesis in the pineal gland. (Modified and supplemented from Alkozi, 2020). Melatonin synthesis is stimulated by signals of darkness coming from the eye through the suprachiasmatic nucleus neurons. Norepinephrine may bind to α- and β-adrenergic receptors on pinealocytes activating both signaling pathways. Adenylyl cyclase is activated to produce cAMP by both adrenergic signaling cascades. cAMP levels regulate AANAT activity, which increase intracellular concentration of NAS converted then to melatonin by HIOMT. 5-HTP, 5-hydroxitryptophan; A, adrenergic receptor; AANAT, arylalkylamine N-acetyltransferase; AC, adenylyl cyclase; HIOMT, hydroxyndole-O-methyltransferase; cAMP, cyclic adenosine monophosphate; CREB, cAMP response element-binding protein; G, G protein; ipRGC, intrinsic photosensitive retinal ganglion cells; NAS, N-acetylserotonin; NE, norepinephrine; NN, adrenergic neurons; PG, pineal gland; PKA, protein kinase A; PKC, C kinase; PLC, phospholipase C; SCN, suprachiasmatic nucleus; and Trp, Tryptophan.

**Figure 2 biomolecules-11-00340-f002:**
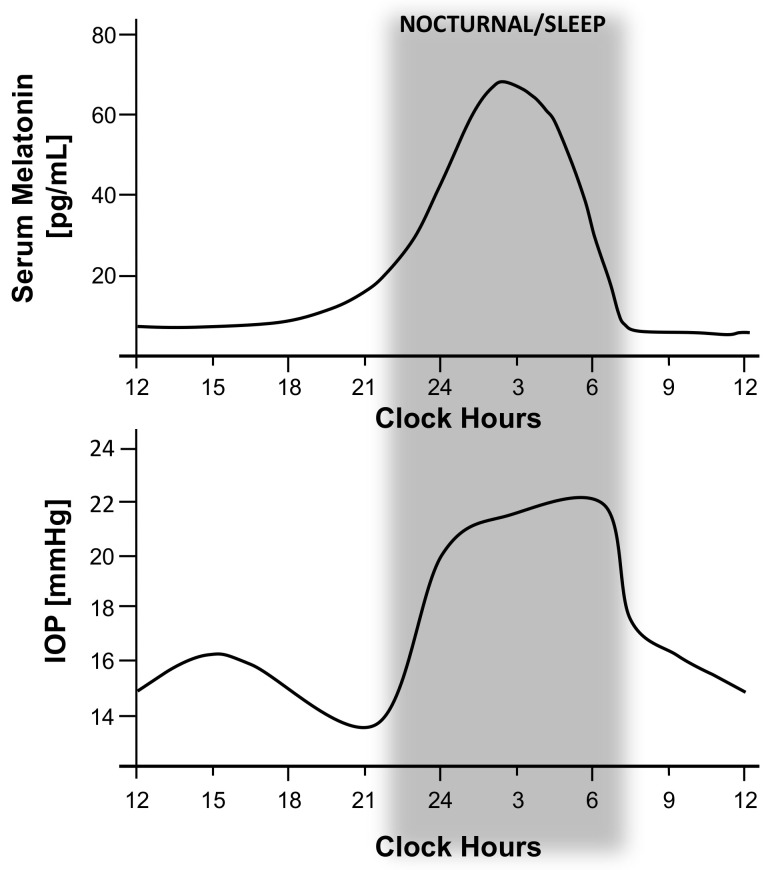
Relationship between increased intraocular pressure (IOP) circadian fluctuation and serum melatonin levels. This picture allows to compare the suggested influence of melatonin levels on IOP circadian fluctuation during the daytime. Graphs show an adapted version of Alkozi, 2020 [186], and Waldhauser, 1988 (above) [187], which represents the 24 h serum melatonin-level pattern in healthy subjects, compared to an adapted version of Grippo, 2013 (below) [188], which shows the 24 h IOP pattern of young healthy subjects. IOP starts to drop about 3 h later after serum melatonin peak appear, which coincides with normal effect of melatonin on IOP decrease.

**Table 1 biomolecules-11-00340-t001:** Summary of principal studies evaluating the effect of melatonin for the treatment of different ocular pathologies.

Ocular Pathology	Reference	Compound	Target	Results
Dry eye	Navarro Gil et al. [79]	Melatonin and analogs (Agomelatine, IIK7, and 5-MCA-NAT)	Rabbits	The topical instillation of analogs of melatonin increased tears secretion: 39% with agomelatine (MT_2_ receptor), 28% with IIK7 (MT_2_ receptor), and 20% with 5-MCA-NAT (unknown receptors). Conversely, there were no changes with the melatonin.
	Hoyle et al. [80]	Ap_4_A + Melatonin	Rabbits	The topical instillation of Ap_4_A increased tear secretion by 10%, while melatonin showed no effect. The synergic effect of both molecules increased tear secretion by 34%.
Corneal wound healing	Crooke et al. [47]	Melatonin and analogs (IIK7 and 5-MCA-NAT)	Rabbits	The topical instillation of melatonin and IIK7 improved rate of corneal healing by around 47%, while 5-MCA-NAT showed no effect. This effect is mediated by the MT_2_ receptor, which is present in corneal epithelial cells.
	Crespo-Moral et al. [81]	Melatonin	Ex vivo Porcine eyeball	The topical exposition of melatonin in an ex vivo corneal-wound model accelerated the healing process. Notably, 60 µg/mL melatonin accelerated the healing rate during 48 h and 90 µg/mL melatonin during 72 h. Unexpectedly, 120 µg/mL melatonin decreased the healing rate after 96 h.
Myopia	Wang et al. [82]	Blue light	Guinea pigs	The retinal stimulation with blue light (480 nm) reduced eye growth, which was related to stimulate synthesis of melanopsin in retina and sclera, and to reduce both synthesis of MT_1_ receptor and production of melatonin in pineal gland.
	Zheng et al. [83]	AA92593 (melanopsin antagonist)	Guinea pigs	The retinal melanopsin inhibition by intravitreal injection of an antagonist AA92593 increased both the eye growth and melatonin levels in the retina, these variables being directly correlated (r = 0.74).
Cataracts	Different authors [84,85,86,87,88,89,90]	Melatonin	Rats	The use of melatonin in rat models reduced lipid peroxidation and promoted both the synthesis of glutathione and antioxidative activity of different enzymes, leading to a reduction in cataract formation.
	Pintor et al. [91]	Yellow light and AA92593	Rabbits	The inhibition of melanopsin present in the lens epithelium with a yellow filter (absorbance between 465-480 nm) and its antagonist AA92593 reduced the concentration of ATP in the aqueous humour.
Retinal damage	Liang et al. [92]	Melatonin	Mice	The muscular injection of melatonin reduced damage of photoreceptors and their apoptosis in a retinal degeneration model.
	Kaur et al. [93]	Melatonin	Rats	The intraperitoneal injection of melatonin reduced levels of VEGF, nitrite, and melatonin in the retina of hypoxic rats, also reducing retinal vascular permeability.
Age-related macular degeneration	Dieguez et al. [94]	Melatonin	Mice	The antioxidant effect of subcutaneous implantation of a pellet of melatonin helped to preserve visual function and retinal structures in a non-exudative AMD model.
	Yi et al. [95]	Melatonin	Humans	In this case series study, the daily oral administration of 3 mg melatonin for 3 months improved the signs of AMD in terms of retinal blood and retinal exudates in more than 90% of patients.

**Table 2 biomolecules-11-00340-t002:** Summary of principal studies evaluating the effect of melatonin and analogs for glaucoma treatment.

Compound	Animal Model/ Patients	Results	Reference
Melatonin	New Zealand white rabbits (NZWR)	22.0% reduction	Pintor et al. [228]
	Hypertensive NZWR	43.6% reduction	Martinez-Águila et al. [207]
	Control (C57BL/6J) and glaucomatous mice (DBA/2J)	19.4 normotensive vs 32.6% glaucomatous mice	Martinez-Águila et al. [236]
	Normotensive patients	17% reduction with 0.5 mg; From 17.9 to 13.8 mmHg (23%) with 10 mg; From 15.7 to 14.7 mmHg (7%) with 1 mg.	Samples et al. [237]; Ismail et al. [209] and Carracedo et al. [210]
Agomelatine	New Zealand white rabbits (NZWR)	20.8% reduction	Martinez-Águila et al. [207]
	Hypertensive NZWR	68.8% reduction	Martinez-Águila et al. [207]
	Glaucomatous patients	From 23.4 to 14.3 mmHg (34%) with 25 mg	Pescosolido et al. [208]
5-MCA-NAT	New Zealand white rabbits (NZWR)	42.5% reduction	Pintor et al. [228]
	Hypertensive NZWR	85.6% reduction	Martinez-Águila et al. [207]
	Control (C57BL/6J) and glaucomatous mice (DBA/2J)	20.7% normotensive vs 29.3% glaucomatous mice. 13% reduction with 3 months treatment vs placebo	Martinez-Águila et al. [236]
	Glaucomatous monkeys (*Macaca fascicularis*)	7.0 mmHg (19%) reduction	Serle et al. [246]
Other melatonin analogs	New Zealand white rabbits (NZWR)	N-Acetyltryptamine 6.0%, 6-chloromelatonin 9.0%, 1-Iodomelatonin 9.0% and 2-phenylmelatonin 8.7% reduction	Pintor et al. [228]
Melatonin + Prazosin	Glaucomatous mice (DBA/2J)	From 16.6 to 9.0 mmHg (46%) reduction	Hanan et al. [244]
5-MCA-NAT + Anhydrases	New Zealand white rabbits (NZWR)	5-MCA-NAT increase dorzolamide effect by 45% vs 32% dorzolamide only for 3 days.	Crooke et al. [229]
Melatonin + adrenergic compounds	New Zealand white rabbits (NZWR)	Melatonin increase brimonidine effect by 29.3% vs 40.6% brimonidine only and 5-MCA-NAT 39.1%, for 4 days. Melatonin increase timolol effect by 39.8% vs 25.8% timolol only and 5-MCA-NAT 42.6% only one day.	Crooke et al. [245]

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
