# Peer review of "Influence of Circadian Rhythm in the Eye: Significance of Melatonin in Glaucoma"

_biomolecules, 2021, doi:10.3390/biom11030340_

Round 1

Reviewer 1 Report

Major concerns

  • Section 1 is verbose, please break this into different subsections, for example, melanopsin, melatonin
  • Line 251: Are there any additional studies on DR? it would be helpful if the relevance of melatonin in DR explained better
  • Please divide section three into subsection for the ease of understanding
  • Considering the focus of the article on glaucoma I will  suggest cutting down discussion on the role of circadian rhythms in other eye diseases, for example, dry eye disease, AMD

Minor concerns

Line 37, 42 correct typo: 'SNC' to SCN

Line 44: keep consistency in naming iPRGC

Line 62: correct Firstly to first

Line 284 is that melatonin? in place of melanin?

et al: Change this to et al. throughout

Author Response

Major concerns

  • Section 1 is verbose, please break this into different subsections, for example, melanopsin, melatonin

Subsections were added.

  • Line 251: Are there any additional studies on DR? it would be helpful if the relevance of melatonin in DR explained better.

Yes, many other studies are reporting the relevance of melatonin in DR due to its effect on retinal inflammation, oxidative stress, angiogenesis and apoptosis (we have summarized some of them). However, this therapeutical effect has only been demonstrated in animal models:

“Several studies performed on animal models demonstrated the protective effect of melatonin against retinal inflammation, oxidative stress, angiogenesis, and apoptosis, which could help to prevent retinal damage associated with diabetes [168,169]. Nevertheless, the clinical benefits of melatonin are still unknown due to the lack of clinical studies.”

  • Please divide section three into subsection for the ease of understanding

Subsections were added.

  • Considering the focus of the article on glaucoma I will  suggest cutting down discussion on the role of circadian rhythms in other eye diseases, for example, dry eye disease, AMD

Sentences were eliminated from lines 177, 183, 187 and 330.

Minor concerns

  • Line 37, 42 correct typo: 'SNC' to SCN. Done
  • Line 44: keep consistency in naming iPRGC. Done
  • Line 62: correct Firstly to first. Done
  • Line 284 is that melatonin? in place of melanin?

Line 284 Melanin is correct. Melatonin would regulate melanin aggregates, but the sentence was eliminated according to previous comment.

  • et al: Change this to et al.  Done

Minor concerns were corrected

Reviewer 2 Report

This is a nice review concerning the significance of melatonin in relationship with glaucoma. I only haver a minor comment:

It is great to see what melatonin can be used to reduct IOP in mice. It will be even nice to additionally review if the melatonin can help slow down vision worsening over time in glaucoma.

Author Response

It is great to see what melatonin can be used to reduct IOP in mice. It will be even nice to additionally review if the melatonin can help slow down vision worsening over time in glaucoma.

A paragraph was added in line 557 reviewing melatonin effect:

“Furthermore, there are some studies which show promising results about a possible slow down vision worsening over time in glaucoma. Park et al. show that RGC survival was greater in retinas treated with melatonin 2 weeks after ischemia-reperfusion [241] and Belforte et al.demonstrate that melatonin prevented and reversed the effect of ocular hypertension on retinal function and diminished the vulnerability of retinal ganglion cells during a six week period [207].”

Reviewer 3 Report

In their article, the authors summarize the findings in regard to ocular diseases and circadian rhythm, and the key molecules related to circadian rhythm. While this paper clearly has its merits and is of high interest, it is sometimes difficult to understand and sometimes lacks precision, which should be addressed before publication. Please find the detailed comments below

Abstract:

The authors mention dopamine in the first sentence of the abstract, but the main body of the text mainly deals with melatonin and melanopsin.

Line 19, 20, the sentence with the two ifs is difficult to understand, please rephrase.

Line 39, generally, light is processed by the photoreceptors and that information relayed by the RGC, with the light sensitive RGCs being an exception. This should be clarified.

Line 65, the lacrimal gland is not part of the eye

Figure 1, how does darkness stimulate ipRGC? Please clarify figure

Myopia

Line 202, what is meant by “control of myopia progression in the age of ocular growth”?

Line 217/218, how is “a constant grow during day and night” circadian?

Line 221, “thanks to previous knowledge” is not a scientific expression

Line 222/223, how can myopic changes be lowered in non-myopic children?

Cataract

Generally, I miss actual connection to circadian rhythm in this chapter. Is there any evidence that it plays a role?

Line 253/255, “…concluded that cataract surgery would help to improve them”, why "would"? Did he show data or was it just an hypothesis?

Retinal diseases

Line 267/268, “autophagy of segments”, this is not correct. Fragments of the segments are shed in a circadian manner and these are phagocytosed by retinal pigment epithelium cells in a circadian manner. This needs to be corrected

Line 292, what is the actual connection of melatonin with angiogenesis?

Generally, I am missing a clear connection to circadian rhythm here. E.g., VEGF is important for AMD development, and VEGF secretion/expression has been implicated to be regulated in a circadian manner.

IOP/glaucoma

Lines 326, why “although”?

Line 332-333, the authors state that in glaucoma patients the IOP ranges between 6 – 15 mmHg, but this is not an increased IOP.

Line 414, the authors state that glaucoma may be the main ophthalmic disease affecting circadian rhythm. Has this been evaluated in other ocular diseases?

Line 415 and following, does the SCN derived melatonin really play an important role in the retina, as the retina can produce its own melatonin?

Melatonin

Why did the authors first describe studies about melatonin analogues and then melatonin itself? Why not start with melatonin?

Line 484 “could be detected in the morning…”, where was the melatonin detected?

A figure overviewing the influence of circadian rhythm, melatonin, etc, on glaucoma would be beneficial.

Conclusion

In general, this is more a summary than a conclusion. Please draw a conclusion in the conclusion.

Line 529/530 “But its role in sleep disorder regulation could be used as a coadjutant treatment of ocular surface diseases” I do not understand the rational of this statement.

Line 540, what is meant by “own circadian rhythm”?

Minor

Line 48, please leave blank between numeral and unit

Author Response

Abstract:

The authors mention dopamine in the first sentence of the abstract, but the main body of the text mainly deals with melatonin and melanopsin.

Dopamine was eliminated from abstract

Line 19, 20, the sentence with the two ifs is difficult to understand, please rephrase.

“to analyze if some treatments could improve their effect on the ocular disease when their posology is established in function of circadian patterns”

Line 39, generally, light is processed by the photoreceptors and that information relayed by the RGC, with the light sensitive RGCs being an exception”. This should be clarified.

Sentence has been changed to a better understanding “Through the eye, light reaches the retina and is processed by retinal ganglion cells (RGCs), which drive whole visual information. But approximately 1%-2% of these cells contain a photopigment denominated melanopsin”.

Line 65, the lacrimal gland is not part of the eye

It was a mistake. Sentences has been corrected. “but currently it is known to be synthesized in many tissues in the body including the eye and ocular annexes”

Figure 1, how does darkness stimulate ipRGC? Please clarify figure

It was an error between melatonin that is stimulated during the night and the photic input that receive ipRGC either in light and darkness. The word stimulation were eliminated to clarify figure.

Myopia

Line 202, what is meant by “control of myopia progression in the age of ocular growth”?

Ocular growth is not constant, as occurs with height, and it is more important during the ages of body development, normally between 7 and 15 years old. “Ocular components during the ages of ocular development. H. Hashemi et al. Acta Ophthalmologica. 2015 Feb;93(1):e74-81.”

Line 217/218, how is “a constant grow during day and night” circadian?

                  It is not related with circadian rhythm but the consequence of blocking it. If the eye is deprived, a constant grow occurs during night and day.

Line 221, “thanks to previous knowledge” is not a scientific expression

Changed for “After that”

Line 222/223, how can myopic changes be lowered in non-myopic children?

                  It is related with the axial increase of the eye that happens during growth in previous non-myopic or myopic children.

Cataract

Generally, I miss actual connection to circadian rhythm in this chapter. Is there any evidence that it plays a role?

Yes, it is. It is related with blue light and how lens turns yellow, blocking stimulation of circadian rhythm. Furthermore, it will be related with sleep disorders affecting mainly to older people.

Line 253/255, “…concluded that cataract surgery would help to improve them”, why "would"? Did he show data or was it just an hypothesis?

                  It was changed to “could”. Paper is a review in which it is stated. “Cataract surgery improved regulation of circadian rhythms measured by the PSQI questionnaire, but the clinical relevance is uncertain”.

Retinal diseases

Line 267/268, “autophagy of segments”, this is not correct. Fragments of the segments are shed in a circadian manner and these are phagocytosed by retinal pigment epithelium cells in a circadian manner. This needs to be corrected.

                  Phrase was changed: “phagocytosis of fragments of these segments by retinal pigment epithelium cells”

Line 292, what is the actual connection of melatonin with angiogenesis?

                  This was a common concern with Reviewer #1. Melatonin would help to prevent retinal angiogenesis as different studies performed on animal models demonstrated. This aspect, among others, was synthesized in the following sentence:

“Several studies performed on animal models demonstrated the protective effect of melatonin against retinal inflammation, oxidative stress, angiogenesis, and apoptosis, which could help to prevent retinal damage associated with diabetes [168,169]. Nevertheless, the clinical benefits of melatonin are still unknown due to the lack of clinical studies.”

Generally, I am missing a clear connection to circadian rhythm here. E.g., VEGF is important for AMD development, and VEGF secretion/expression has been implicated to be regulated in a circadian manner.

                  We have included the relationship between the levels of VEGF secreted by RPE with the circadian clock in the first paragraph of Retinal diseases section:

“On the other hand, the levels of vascular endothelial growth factor (VEGF) secreted by retinal pigment epithelium regulating the angiogenesis are increased under daylight conditions [4].” has been added.

IOP/glaucoma

Lines 326, why “although”?

                  It was an error. It was changed to “nevertheless”

Line 332-333, the authors state that in glaucoma patients the IOP ranges between 6 – 15 mmHg, but this is not an increased IOP.

                  It was changed to: “IOP fluctuates in a range between 6 and 15 mmHg”

Line 414, the authors state that glaucoma may be the main ophthalmic disease affecting circadian rhythm. Has this been evaluated in other ocular diseases?

                  Not completely, but based in the information provided, it is the one that fluctuates more during the day and thus it is related with circadian rhythm.

Line 415 and following, does the SCN derived melatonin really play an important role in the retina, as the retina can produce its own melatonin?

                  Yes, it does. As it is stated in this section, IOP is also influenced by OPP which is regulated by main circadian rhythm from SCN derived melatonin, not by retinal melatonin.

Melatonin

Why did the authors first describe studies about melatonin analogues and then melatonin itself? Why not start with melatonin?

                  Melatonin is described in the first line of the section 4 indicating its hypotensive effect.

Line 484 “could be detected in the morning…”, where was the melatonin detected?

A figure overviewing the influence of circadian rhythm, melatonin, etc, on glaucoma would be beneficial.

                  It is referring to the effect of melatonin, not melatonin itself. Sentence was changed to: “showing that melatonin not only reduced IOP in an acute effect but could also reduce IOP in the morning during the 3 days tested”

Conclusion

In general, this is more a summary than a conclusion. Please draw a conclusion in the conclusion.

                  It was added:       

“6. Conclusion

The diagnosis of certain eye conditions could be improved if an all-day monitoring of some parameters were made. It is currently possible to analyze the IOP using a specific contact lens sensor able to capture the IOP for 24 hours [253]. This type of measurement could help to get better diagnosis and therefore to prescribe the best treatment for the patient. Regarding treatments, two ways should be studied; the first one, to analyze if some treatments could improve their effect on the ocular disease if their posology could be established in function of circadian patterns. For instance, is it better to instill atropine for myopia control progression or a hypotensive drug for glaucoma at the morning or at night? The second one, to evaluate new drugs to treat eye pathologies related to the circadian rhythm, as melatonin or its analogues, not only as the main treatment but as coadjutant, improving the circadian pattern or for its antioxidant and antiangiogenic properties.”

Line 529/530 “But its role in sleep disorder regulation could be used as a coadjutant treatment of ocular surface diseases” I do not understand the rational of this statement.

It is related with sleep disorders as mentioned before, but the sentence were modified: “But its role in sleep disorder regulation could be used as coadjutant treatment of ocular surface diseases due to the improvement in sleep disorders”

Line 540, what is meant by “own circadian rhythm”?

It means that circadian rhythm modulates axial growth.

Minor

Line 48, please leave blank between numeral and unit. Done

Round 2

Reviewer 1 Report

The authors have addressed comments raised by this reviewer

Author Response

Thank you so much for  your time and effort to improve this review.
